# Effect of Fermentation with *Streptococcus thermophilus* Strains on In Vitro Gastro-Intestinal Digestion of Whey Protein Concentrates

**DOI:** 10.3390/microorganisms11071742

**Published:** 2023-07-03

**Authors:** Ahmed Helal, Sara Pierri, Davide Tagliazucchi, Lisa Solieri

**Affiliations:** 1Department of Food and Dairy Sciences and Technology, Damanhour University, Damanhour 22516, Egypt; ahmed.helal@damanhour.edu.eg; 2Department of Life Sciences, University of Modena and Reggio Emilia, Via Amendola, 2-Pad. Besta, 42100 Reggio Emilia, Italy; 277306@studenti.unimore.it (S.P.); davide.tagliazucchi@unimore.it (D.T.); 3National Biodiversity Future Center (NBFC), 90133 Palermo, Italy

**Keywords:** *Streptococcus thermophilus*, whey protein concentrate, bioactive peptides, peptidomics, gastro-intestinal digestion, angiotensin-converting enzyme inhibition, valine–proline–proline, isoleucine–proline–proline

## Abstract

Three *Streptococcus thermophilus* strains, namely RBC6, RBC20, and RBN16, were proven to release bioactive peptides during whey protein concentrate (WPC) fermentation, resulting in WPC hydrolysates with biological activities. However, these bioactive peptides can break down during gastro-intestinal digestion (GID), hindering the health-promoting effect of fermented WPC hydrolysates in vivo. In this work, the effect of simulated GID on three WPC hydrolysates fermented with *S. thermophilus* strains, as well as on unfermented WPC was studied in terms of protein hydrolysis, biological activities, and peptidomics profiles, respectively. In general, WPC fermentation enhanced protein hydrolysis compared to unfermented WPC. After in vitro GID, WPC fermented with *S. thermophilus* RBC20 showed the highest antioxidant activity, whereas WPC fermented with strain RBC06 displayed the highest angiotensin-converting enzyme (ACE)- and dipeptidyl peptidase IV (DPP-IV)-inhibitory activities. Peptidomics analysis revealed that all digested WPC samples were highly similar to each other in peptide profiles, and 85% of the 46 identified bioactive peptides were shared among fermented and unfermented samples. However, semi-quantitative analysis linked the observed differences in biological activities among the samples to differences in the amount of bioactive peptides. The anti-hypertensive peptides VPP and IPP, as well as the DPP-IV-inhibitory peptide APFPE, were quantified. In conclusion, WPC fermentation with *S. thermophilus* positively impacted protein hydrolysis and bioactive peptide release during GID.

## 1. Introduction

Whey is the main by-product of the dairy industry, representing the residual yellowish and opaque liquid emerging from the first step of cheese manufacturing [1]. The cheese whey production has been estimated to increase from year to year due to the constant growth of the dairy sector, driven by the increased demand in cheese [2]. This trend poses important environmental issues, as whey is considered the most important environmental pollutant of the dairy sector due to the high chemical and biological oxygen demands [1,2]. For the same reason, while cheese whey is considered pollutant, it can also be viewed as reserve for compounds with a high nutritional value that make whey exploitation attractive for several industrial sectors [2,3,4]. In the last years, efforts have been made to support sustainable cheese whey valorization and management [1,5]. Within the modern concept of integrated biorefinery, cheese whey has been exploited to develop value-added products, such as whey powder, whey permeate, whey protein isolate, whey protein concentrate (WPC), whey-based beverages, bioethanol, biohydrogen, protein hydrolysates, and organic acids [1,2,5,6].

The use of cheese whey or whey protein preparations (such as WPC) to produce functional whey-based beverages is one of the most economical and attractive applications in the field of human nutrition and health [1,2,3]. Whey proteins exhibit high nutritional value due to the presence of essential amino acids, such as cysteine, methionine, and branched-chain amino acids [4]. In addition, whey proteins have a plethora of biological activities, when bioactive peptides encrypted in their sequences are released through proteolytic breakdown [7].

Bioactive peptides can be defined as short amino acid sequences that may have important impacts on human health, such as anti-hypertensive and angiotensin-converting enzyme (ACE)-inhibitory activity, anti-diabetic properties, immunomodulatory and anti-inflammatory activities, as well as anti-cancer and antioxidant activities [8,9,10]. Numerous bioactive peptides have been identified until now through the hydrolysis of proteins from diverse sources (animal, marine, or plant proteins). Undoubtedly, milk proteins such as caseins and whey proteins are the most investigated sources of bioactive peptides [7,8].

Peptidomics is an omics-based technique that was developed at the beginning of the 21st century from the shot-gun proteomics approach. The goal of peptidomics is to identify the pool of peptides present in a biological sample, including foods. The the field of food science peptidomics is now widely applied for the identification and quantification of bioactive peptides, as well as in biomarker discovery for food authentication [11]. The peptidomics workflow includes peptide extraction and purification, separation via liquid chromatography, followed by tandem mass spectrometry experiments, and bioinformatics analysis for peptide identification and quantification. Liquid chromatography coupled with mass spectrometry analysis and bioinformatics tools (such as peptide sequencing database, semi-quantitative analysis database, and bioactive peptides database) is the technique of choice for peptidomics studies, able to accomplish high-throughput bioactive peptide characterization [12].

Bioactive peptides can be already present in a specific food (such as fermented dairy products like cheese, yogurt, and fermented milk) or may be generated following the gastro-intestinal digestion (GID) of food proteins [8,13]. Nevertheless, whey proteins, especially β-lactoglobulin, are characterized by a rigid structure that hamper their gastric hydrolysis, slowing down the intestinal degradation and the release of bioactive peptides during digestion [14]. As a result, whey proteins are degraded during GID more slowly than caseins. For example, Dupont et al. found that 72% of the initial amount of β-lactoglobulin was still present after 30 min of intestinal digestion [15], whereas Picariello et al. identified residual intact β-lactoglobulin after 120 min of intestinal digestion [16]. Therefore, preliminary hydrolysis of whey proteins could be of paramount importance to foster whey protein hydrolysis and the release of bioactive peptides [6,17].

Besides enzymatic hydrolysis catalyzed by purified proteases, lactic acid fermentation is a sustainable and economical alternative to release bioactive peptides from whey proteins, resulting in whey hydrolysates with biological activities [2]. For example, Mazorra-Mazzano et al. found that lactic acid bacteria (LAB) release peptides with ACE-inhibitory activity during cheese whey fermentation [18]. Similarly, *Lactobacillus helveticus* and *Lactobacillus acidophilus* produced several bioactive peptides during the fermentation of a whey protein isolate [19]. Indeed, WPC fermentation with either *Lactobacillus helveticus* or *Streptococcus thermophilus* improved the content of bioactive peptides and enhanced the resulting biological activities [20,21]. However, the structure of peptides, and consequently their activity, can be modified in vivo by GID, as gastro-intestinal proteases and stressed conditions can degrade and inactivate WPC peptides. Therefore, the health effects of bioactive peptides are greatly influenced by their digestive stability. Indeed, during GID, bioactive peptides already present in foods may be degraded or new bioactive peptides can be released from oligopeptide sequences through the action of pepsin and intestinal proteases [8,11]. However, no studies have been carried out until now to evaluate the impact of GID on the stability and release of bioactive peptides in fermented whey hydrolysates.

In a previous study, WPC was successfully fermented and fortified in bioactive peptides using selected *S. thermophilus* strains isolated from Parmigiano Reggiano cheese natural whey starter, namely RBC06, RBC20, and RBN16 [21]. Here, these three fermented WPC hydrolysates, as well as unfermented WPC, were submitted to simulated GID to establish how their protein hydrolysis degree, bioactive peptide profiles, and biological activities change after in vitro GID when analyzed within a comparative framework.

## 2. Materials and Methods

### 2.1. Materials

Chemicals and reagents for in vitro digestion, enzymatic assays, and antioxidant activity analysis were from Sigma-Aldrich (Milan, Italy), whereas the solvents for mass spectrometry analysis were supplied by Biorad (Hercules, CA, USA). Medium M17 for streptococci growth and lactose were supplied by Oxoid (Basingstoke, Hampshire, UK). Whey protein concentrate (WPC80) containing 81% (*w*/*w*) protein and 4.5% (*w*/*w*) fat was purchased from a local producer (Reire srl, Reggio Emilia, Italy).

The ultra-filtration units (Amicon Ultra-0.5 regenerated cellulose filters; cut-off of 10 kDa) were purchased from Millipore (Milan, Italy). Standard peptides for quantitative analysis (APFPE, IPP and VPP; 99% purity) were synthesized by Bio-Fab Research (Rome, Italy).

### 2.2. Whey Protein Concentrate Preparation

The whey protein concentrate (WPC) solution was formulated by dissolving 15 g of WPC80 and 50 g of lactose in 1 L of distilled water. The complete dissolution of the powders was reached by stirring at 50 °C for 2 h. After that, the preparation was heat-treated at 95 °C for 10 min [21], immediately cooled in ice-cold water, and finally stored at −20 °C until use.

### 2.3. Microbial Cultures

Three different strains of *S. thermophilus*, namely RBC06, RBC20, and RBN16, were formerly isolated from natural whey starters used in Parmigiano Reggiano cheese production and selected for their ability to ferment WPC and release bioactive peptides [21,22]. Twenty-four-hour-old cells cultured on an M17 medium at 42 °C were harvested via centrifugation (9000× *g* for 15 min at 4 °C), washed twice with a sterile saline solution (9 g/L NaCl), and re-suspended in an aliquot of saline solution at the cell density of log 9.0 cfu/mL, before being used for subsequent fermentation trials.

### 2.4. Fermentation and In Vitro Gastro-Intestinal Digestion

Fermentation trials were carried out in triplicate, as fully described in Solieri et al. [21]. Briefly, 10 mL aliquots of a reconstituted WPC medium pre-conditioned at 42 °C in a water bath were inoculated with each strain (1% *v*/*v*) and incubated at 42 °C for 72 h. In parallel, a negative control was performed by using non-inoculated (unfermented) WPC, treated exactly as the fermented WPC (i.e., same incubation time and temperature). At the end of each fermentation trial, the three replicates were pooled together and subjected to in vitro gastro-intestinal digestion following the INFOGEST protocol (an international network of excellence on the fate of food in the gastro-intestinal tract) [23]. Briefly, 1 mL of fermented or unfermented WPC was added to 1 mL of simulated salivary fluid and salivary α-amylase (150 U/mL), and subsequently incubated at 37 °C for 2 min in a rotating wheel (10 rpm). To simulate the gastric phase of digestion, gastric fluid was added in the amount of 5 mL, and after bringing the pH to 3 with 6 mol/L HCl, the gastric protease pepsin was added to achieve a final concentration of 2000 U/mL. The bolus was then incubated for 120 min at 37 °C under rotation (10 rpm). The further intestinal step of the digestion was initiated by adding 4 mL of intestinal fluid, which was followed by the raising of the pH to 7.5. After 30 min of incubation at 37 °C, pancreatin was added so that the final concentration of trypsin was 100 U/mL. After 120 min of incubation at 37 °C under rotation (10 rpm), the digested sample was boiled for 5 min to inactivate the proteases and centrifuged for 20 min at 4 °C and 10,000× *g.*

Control digestion was carried out by substituting WPC with water to consider the possible interferences due to the digestive system in the applied assays.

Digestions were carried out in triplicate for each fermentation trial and the three replicated digestions were then pooled together before analysis.

The composition of the digestive fluids and the full protocol were reported in Brodkorb et al. [23].

### 2.5. Assessment of Protein Hydrolysis

Protein hydrolysis was determined by quantifying the total free amino groups in digested samples through the TNBS assay, as previously reported [24]. Preliminarily, 100 μL of each in-vitro-digested sample was mixed with 20 μL of 50% trichloroacetic acid (TCA) and incubated at room temperature for 10 min to precipitate undigested proteins. Undigested proteins were removed via centrifugation at 10,000× *g* for 20 min at 4 °C, before the protein hydrolysis assay. The results were expressed as mmol of leucine equivalents/L of WPC. The data were corrected by considering the contribution of the control digestion.

### 2.6. Biological Activity Assays

Low-molecular-weight peptide fractions were extracted from the in-vitro-digested samples via ultrafiltration at a 10 kDa cut-off, as previously described [25].

The ABTS (2,2-azino-bis(3-ethylbenzothiazoline-6-sulphonic acid)) assay was applied to evaluate the radical scavenging activity of low-molecular-weight peptide fractions [26]. The results were expressed as mg of ascorbic acid/mmol of peptides.

The ability of low-molecular-weight peptide fractions to inhibit the activity of the angiotensin-converting enzyme (ACE) was assessed using the tripeptide N-[3-(2-furyl)acryloyl]-L-phenylalanylglycyl-glycine (FAPGG) as a substrate and following the procedure reported in Solieri et al. [21]; whereas the ability to inhibit the enzyme dipeptidyl peptidase IV (DPP-IV) was determined using the dipeptide glycine–proline–p-nitroanilide (Gly–Pro–pNA) as a substrate and following the protocol reported in Tagliazucchi et al. [27].

The data for both enzymatic assays were reported as IC_50_ values, expressed as μmol of peptides/mL. The IC_50_ values (defined as the ability of the sample to inhibit 50% of enzymatic activity) were computed by non-linear regression analysis and plotting the percentage of enzyme inhibition versus the base-10 logarithm of the peptide concentration in the sample. The percentage of enzyme inhibition was calculated by carrying out the assays in presence of different concentrations of low-molecular-weight peptides.

### 2.7. High-Resolution Mass Spectrometry Analysis of the Peptide Profiles of Low-Molecular-Weight Peptide Fractions Extracted from In-Vitro-Digested Samples

The peptide profiles of the low-molecular-weight peptide fractions were achieved through high-resolution mass spectrometry analysis using a Q Exactive Hybrid Quadrupole-Orbitrap Mass Spectrometer (Thermo Scientific, San Jose, CA, USA). Chromatographic separation was performed with a UHPLC (UHPLC Ultimate 3000 separation module, Thermo Scientific, San Jose, CA, USA) module equipped with a C18 column (Acquity UPLC HSS C18 reverse phase, 2.1 mm × 100 mm, 1.8 µm particle size, Waters, Milan, Italy). The full description of the chromatographic and mass spectrometry conditions is reported in Martini et al. [28]. Tandem mass spectrometry analysis (MS/MS) was carried out by using data-dependent acquisition. Peptide sequencing was carried out using Mascot software (version 2.7.0; release date January 2020) and applying the same parameters previously described [28]. Only the peptides identified with a significance threshold of *p* < 0.05 were included. Each digested sample was injected in triplicate.

### 2.8. Peptidomics Analysis and Label-Free MS Peak Quantification

Peptidomics analysis was performed using Skyline software (version 22.2; release date September 2022) [29], following the procedure reported in Dallas and Nielsen [30]. First, the list of peptides identified with Mascot software within different samples were saved as a .dat file and used to create a specific library in Skyline software. Then, the full-scan mass spectral data obtained from different samples were processed for the MS peak quantification of each identified peptide [28]. The obtained data were then filtered and the peaks that did not meet the criteria or were too close to the noise level to be visually discernible were excluded from the dataset. The criteria used were a mass tolerance ≤5 ppm and an isotope scalar product score (idotp) ≥ 80. The peak area values of identical peptides, but with different modifications (such as different protonation pattern, methionine oxidation and glutamine/asparagine deamidation), were summed. Only peptides belonging to major milk proteins (β-casein, αS1-casein, αS2-casein, κ-casein, β-lactoglobulin, and α-lactalbumin) were considered.

### 2.9. Identification and Relative Quantification of Bioactive Peptides

The identification of bioactive peptides was carried out using the Milk Bioactive Peptide Database (MBPDB, http://mbpdb.nws.oregonstate.edu/, accessed on 30 January 2023) [31]. Only peptides that shared 100% of homology with previously reported bioactive peptides were included in the list. The relative quantification of identified ACE-inhibitory and DPP-IV-inhibitory peptides was performed using the Skyline dataset as reported above.

### 2.10. Quantification of the Bioactive Peptides VPP, IPP, and APFPE

The ACE-inhibitory peptides IPP and VPP, as well as the DPP-IV-inhibitory peptide APFPE, were quantified using synthesize peptides (purity ≥ 99%), following the parallel reaction monitoring procedure already described [32].

### 2.11. Statistical Analysis

Data were reported as the mean ± standard deviation (SD) for triplicate experiments. A one-way ANOVA followed by a Tukey post hoc test was used for statistical analysis through GraphPad Prism 6.0 (GraphPad Software, San Diego, CA, USA). The normal distribution of the data was checked using the Shapiro–Wilk test. Differences were considered significant when *p* < 0.05.

## 3. Results

### 3.1. Whey Protein Concentrate Fermentation Improved Protein Hydrolysis after In Vitro Gastro-Intestinal Digestion

The selected *S. thermophilus* strains RBC06, RBC20, and RBN16 fermented WPC and hydrolyzed WPC proteins thanks to their proteolytic activity (Figure 1A). Strain RBN16 overcame strains RBC06 and RBC20 in proteolytic capacity, releasing a significantly higher amount (*p* < 0.05) of peptides after 72 h of fermentation. No significant differences (*p* > 0.05) were found between RBC06 and RBC20 strains in proteolytic ability. These results agreed with a recent study [21]. Proteolytic strains of *S. thermophilus* are equipped with a complex and peculiar proteolytic system that is required for their growth in an amino-acid-free medium such as milk [33]. The first step in protein hydrolysis is mediated by the action of extracellular cell-envelope proteases (known as PrtS in *S. thermophilus*) that can cleave caseins (and to a lesser extent whey proteins) in oligopeptides of different sizes [33]. Next, the oligopeptides are transported in the cell where an array of different peptidases, some of them specific to *S. thermophilus*, cleave the oligopeptides in short peptides and free amino acids [8,33].

Then, unfermented and fermented WPC samples were subjected to in vitro gastro-intestinal digestion. As reported in Figure 1B, protein hydrolysis after in vitro digestion was significantly higher (*p* < 0.05) in fermented WPC samples with respect to the unfermented samples. In unfermented WPC, the amount of free amino groups increased from 2.51 ± 0.19 mmol of leucine equivalent/L of WPC in undigested sample, to 35.85 ± 0.41 mmol of leucine equivalent/L of WPC after in vitro digestion. The amount of free amino groups in fermented WPC samples was 2.7 to 3.7 higher than that observed in unfermented WPC after in vitro digestion. Among the fermented samples, protein hydrolysis after in vitro digestion was not significantly different (*p* > 0.05) between WPC fermented with RBC20 and RBN16 strains, whereas the amount of free amino groups released after in vitro digestion of WPC fermented with the RBC06 strain was the lowest (*p* < 0.05). Therefore, gastro-intestinal proteases were also able to hydrolyze WPC proteins in the unfermented sample, but with less effectiveness in comparison with the fermented ones. Nevertheless, results showed that WPC fermentation greatly enhanced the hydrolysis of WPC proteins during GID.

Previous studies suggested that milk fermentation by LAB may improve the following proteolysis during GID [34]. However, no studies have been carried out until now on the effect of fermentation on whey protein digestion. Fermentation promotes whey protein denaturation and hydrolysis, resulting in the release of oligopeptides, of which the peptide bonds are more easily hydrolysed by gastric and intestinal proteases during digestion.

### 3.2. Biological Activity Analysis of Low-Molecular-Weight Peptide Fractions from In-Vitro-Digested Fermented and Unfermented Whey Protein Concentrates

Biological activity analysis was carried out on low-molecular-weight peptide fractions (<10 kDa) extracted from in-vitro-digested fermented and unfermented WPC samples, and the results are displayed in Figure 2.

All peptide fractions showed ABTS radical scavenging activity, with some differences (Figure 2). In-vitro-digested WPC, unfermented and fermented with RBC20, exhibited the highest radical scavenging activity (171.81 ± 7.07 and 163.76 ± 6.93 mg of ascorbic acid/L of WPC), followed by in-vitro-digested WPC fermented with RBN16 (127.96 ± 6.71 mg of ascorbic acid/L of WPC) and RBC06 (93.06 ± 7.31 mg of ascorbic acid/L of WPC). Accordingly, Bustamante et al. [17] found that the antioxidant activity of WPC increased after in vitro gastro-intestinal digestion and was higher than that observed in in-vitro-digested WPC hydrolyzed with flavourzyme. On the contrary, Power-Grant et al. [35] found higher antioxidant activity in whey protein hydrolysates compared to the WPC after gastro-intestinal digestion.

Different results were obtained for the ACE inhibitory assay (Figure 2). The highest inhibitory ability was found for the peptide fraction extracted from in-vitro-digested WPC fermented with RBC06 (0.256 ± 0.011 μmol of peptides/mL), followed by unfermented WPC (0.547 ± 0.032 μmol of peptides/mL). No significant differences (*p* > 0.05) were found in ACE-inhibitory activity between the peptide fractions extracted from in-vitro-digested WPC fermented with RBC20 and RBN16.

Peptide fraction from RBC06 fermented and digested WPC also displayed the lowest IC_50_ value against the DPP-IV enzyme (0.281 ± 0.012 μmol of peptides/mL). In this case, the second most potent sample was in-vitro-digested WPC fermented with RBC20 (0.583 ± 0.047 μmol of peptides/mL), although there were no significant differences (*p* > 0.05) with unfermented WPC (0.613 ± 0.048 μmol of peptides/mL). Once again, in-vitro-digested WPC fermented with RBN16 exhibited the lowest inhibitory activity (Figure 2).

In general, we did not find any direct correlation between levels of biological activities and degrees of protein hydrolysis; however, the results suggested that a lower hydrolysis degree (as observed in in vitro digested WPC fermented with RBC06) may result in higher ACE and DPP-IV inhibitory activities.

A comparison of the ACE- and DPP-IV-inhibitory activity values of fermented WPC before and after simulated GID revealed significant differences between digested and undigested samples [21]. In the case of ACE-inhibitory activity, an increase in the IC_50_ values was observed after the digestion of WPC fermented with RBC20 and RBN16 strains; whereas, in the case of WPC fermented with RBC06, in vitro digestion decreased the IC_50_ value against ACE [21]. Regarding the DPP-IV-inhibitory activity, a decrease in the IC_50_ values was observed after in vitro GID of all samples [21]. All results suggested that there was not a correlation between the protein hydrolysis degree and the biological activity of peptides. Rather, the biological activity profile is dependent on a fine balance between the release and degradation of bioactive peptides during protein hydrolysis. Previous studies have already highlighted that biological activities may decrease or increase as protein hydrolysis proceeds [17,25,35,36].

### 3.3. Peptidomics Profile of In-Vitro-Digested Fermented and Unfermented WPCs

Peptidomics profiles of the low-molecular-weight peptide fractions from in-vitro-digested fermented and unfermented WPCs were analyzed via high-resolution mass spectrometry, Mascot software for peptides identification, and Skyline software for semi-quantitative analysis. The Mascot and Skyline outputs for the different samples are displayed in Appendix A.

The number of identified peptides was quite similar for all the samples (Figure 3A), ranging from 484 peptides identified in in-vitro-digested WPC fermented with RBN16, to 456 peptides identified in in-vitro-digested unfermented WPC. Most identified peptides originated from β-casein, followed by αS1-casein and κ-casein (Figure 3A). Only a minor portion of peptides (ranging from 19.0% to 21.1%) was released from major whey proteins, β-lactoglobulin and α-lactalbumin. Milk caseins are more efficiently hydrolyzed in the gastro-intestinal tract, compared to whey proteins that show some resistance to the action of gastro-intestinal proteases and are degraded more slowly than caseins, mainly due to their rigid and highly ordered structure in contrast to the un-ordered and flexible structure of caseins [15,16,37,38,39]. Moreover, caseins are the preferred substrates for LAB cell-envelope proteases, and during fermentation, they are better hydrolyzed than whey-protein-releasing peptides which are more susceptible to the following gastro-intestinal hydrolysis [21,40].

The occurrence of caseins in WPC preparation (which is supposed to contain only whey proteins) may be due to the presence of residual caseins, non-precipitated during cheese-making, or the formation of water-soluble casein peptides released from the parent proteins during the cheese production process [41]. Previous studies carried out with whey protein or original cheese whey predominantly identified peptides derived from casein hydrolysis during fermentation [19,21,42,43].

A comparison of the peptide profiles of the different samples (Figure 3B) revealed that 419 peptides (corresponding to the 83.8% of the total identified peptides) were commonly released after in vitro digestion of the fermented and unfermented WPC samples. In our previous work, we characterized a large diversification in the peptide profiles of WPC fermented with strains RBN16, RBC20, and RBC06 [21]. Despite this diversity, the broad substrate specificity of gastro-intestinal proteases evened out the differences in peptide profiles, so that at the end of the digestion, the samples were characterized by a great similarity in the released peptides.

Despite the high similarity of the different samples in peptide profiles, semi-quantitative analysis revealed significant differences among the sum of the intensity (peptide abundance measured as the area under the peak for each specific peptide) of the identified peptides in in-vitro-digested WPC samples (Figure 3C). The highest peptide abundance was found for the digested unfermented WPC followed by digested WPC fermented with *S. thermophilus* RBN16, RBC06, and RBC20, respectively. These results disagree with those reported in Figure 1, where digested unfermented WPC was characterized by the lowest level of proteolysis. The high degree of hydrolysis recorded for in-vitro-digested fermented WPC samples may indicate that released peptides were broken down to single amino acids or short di-peptides (not detectable with the current peptidomics approach), resulting in a high level of proteolysis and lower abundance of peptides.

The analysis of the peptide abundance by protein (Figure 3C) revealed that the highest peptide abundance was found for β-casein, followed by αS1-casein, confirming their great predisposition to gastro-intestinal hydrolysis. Instead, the lowest peptide abundance was found for α-lactalbumin, probably due to its low concentration and high resistance to hydrolysis. Regarding β-lactoglobulin, the highest peptide abundance was found in digested WPC fermented with RBC06, suggesting that WPC fermentation with this strain resulted in a better hydrolysis of β-lactoglobulin during gastro-intestinal digestion.

### 3.4. Bioactive Peptide Identification and Correlation with Biological Activity Profiles

A total of 46 peptides with 100% sequence homology with known bioactive peptides were identified (Table 1). Most of these peptides showed ACE-inhibitory activity (27 peptides), DPP-IV-inhibitory activity (11 peptides), and antioxidant activity (9 peptides). An additional six peptides exhibited anti-microbial activity, whereas four peptides were immunomodulatory, and four peptides were anti-inflammatory. Some other peptides displayed anti-cancer, opioid, or anxiolytic activities.

Once again, most of the identified bioactive peptides (39 out of 46) were found in all the samples (Table 1).

#### 3.4.1. ACE-Inhibitory Peptides

A total of 27 peptides with previously reported ACE-inhibitory activity were identified in the digested WPC samples (Table 1). Some of the identified peptides were displayed in vivo anti-hypertensive activity. In particular, the lactotripeptides IPP and VPP were able to reduce blood pressure by about 4 mmHg in human volunteers at doses ranging between 3 to 10 mg/day, as well as in spontaneously hypertensive rats, with a recorded decrease in systolic blood pressure of 18 and 20 mmHg [44,45,46,47]. Additional peptides with demonstrated in vivo anti-hypertensive activity were β-casein-derived hexapeptide LHLPLP, the pentapeptide HLPLP, and their derived fragments LPLP and PLP that were able to significantly reduce systolic blood pressure (from 16 to 25 mmHg) in SHR [48,49]. Moreover, the peptide KVLPVPQ released from β-casein decreased blood pressure by 31.5 mmHg in SHR, despite its high IC_50_ value against ACE [50].

Some peptides also exhibited low IC_50_ values against the enzyme ACE (Table 1). The lowest IC_50_ values of 3, 5, and 8 μmol/L were found for peptides LHLPLP, IPP, and PP, respectively.

Finally, further three β-casein-derived peptides (NLHLPLP, VLPVPQK, and IQA) displayed IC_50_ values between 15 and 51 μmol/L [51,52,53].

Since 88.9% of ACE-inhibitory peptides (24 out of 27 ACE-inhibitory peptides) were present in all samples, simply counting the number of ACE-inhibitory peptides is not enough to explain the different ACE-inhibitory activities observed among the digested WPC samples.

Therefore, to try to get more information on the relationship between the ACE-inhibitory activity and the identified peptides, semi-quantitative analysis was performed. In the first set of analyses, the abundance (determined as the area under the peak) of each ACE-inhibitory peptide was summed and then normalized for the peptide concentration in the specific sample. In the second set of analyses, to consider the peptide inhibitory potency, the abundance of each peptide was divided by the respective IC_50_ value and then normalized for the peptide concentration in the specific sample. Data reported in Figure 4 revealed a strong relationship between the IC_50_ values against ACE and the ACE-inhibitory peptide abundance (Pearson r coefficient of −0.9479). In-vitro-digested WPC fermented with RBC06 displayed the highest abundance of ACE-inhibitory peptides, in agreement with the highest ACE-inhibitory activity (e.g., the lowest IC_50_ value) (Figure 4A). Some of the most potent ACE-inhibitory peptides, such as IPP, VPP, NLHLPLP, VLPVPQK, and IQA, were found at the highest amount in in-vitro-digested WPC fermented with RBC06. Similarly, a strong relationship was also found when the peptide inhibitory potency was considered (Pearson r coefficient of −0.9338) (Figure 4B). All together, these data pointed out that the identified bioactive peptides were mainly responsible for the ACE-inhibitory activity of the low-molecular-weight fractions extracted from the digested WPC samples.

#### 3.4.2. DPP-IV-Inhibitory Peptides

A total of 11 peptides with previously reported DPP-IV-inhibitory activity (see Table 1 for peptide sequences and IC_50_ values) were detected within the analyzed samples, and 10 of these were identified within all WPC samples. Four peptides (β-casein-derived peptides LPVPQ and VPYPQ, αS1-casein-derived peptide APFPE, and β-lactoglobulin-derived peptide IPA) showed IC_50_ values against DPP-IV lower than 50 μmol/L [29,54,55].

The semi-quantitative analysis on the abundance of DPP-IV-inhibitory peptides did not reveal a direct relationship (Pearson r coefficient of −0.4115) between the IC_50_ values against DPP-IV determined in WPC samples and the relative amount of these peptides (Figure 4C). However, when the peptide inhibitory potency was considered, a better, but still a not significant, relationship (Pearson r coefficient of −0.8049) was observed (Figure 4D). In-vitro-digested WPC fermented with RBC06 that showed the highest DPP-IV-inhibitory activity also displayed the greatest concentration of peptides, with the lowest IC_50_ value (especially VPYPQ, APFPE, and IPA). Moreover, the non-significant relationship could suggest that other not-yet-identified DPP-IV-inhibitory peptides were present in the digested WPC samples.

#### 3.4.3. Antioxidant Peptides

As depicted in Table 1, nine peptides with already-demonstrated antioxidant properties were identified in in-vitro-digested WPCs. Eight out of nine peptides were found in all in-vitro-digested WPC samples. Some of these peptides were previously characterized for their ABTS radical scavenging activity, such as β-casein-derived ABTS scavenging peptides, VPYPQ and PYPQ, identified within all samples, and αS1-casein-derived peptide YLG, only identified in in-vitro-digested unfermented WPC and WPC fermented with RBN16 [56,57,58].

### 3.5. Quantification of Bioactive Peptides VPP, IPP, and APFPE

The two anti-hypertensive tripeptides VPP and IPP were quantified in the low-molecular-weight peptide fractions obtained from in-vitro-digested WPCs. As shown in Table 2, the highest amount of both tripeptides was detected in in-vitro-digested WPC fermented with RBC06, whereas the lowest one was revealed in in-vitro-digested unfermented WPC. The data suggested that WPC fermentation with *S. thermophilus* strains maximizes the release of the anti-hypertensive peptides VPP and IPP during in vitro digestion. Within all samples, the peptide IPP was released at higher amounts than peptide VPP (*p* < 0.05). Contrariwise, after WPC fermentation with *S. thermophilus* strains RBC06, RBC20, and RBC16, the tripeptide VPP was found at higher amount than IPP [21]. These results suggest that while LAB cell-envelope proteases are more efficient in releasing VPP from β-casein, digestive proteases are more effective in releasing the tripeptide IPP, as already observed during the in vitro digestion of bovine milk [59].

Since peptides VPP and IPP were active in vivo at doses ranging between 3 and 10 mg/day, based on the amount of these tripeptides released after in vitro digestion (Table 2), it is possible to speculate that a portion of 100 mL of WPC fermented with *S. thermophilus* strain RBC06 may result in an in vivo effect in hypertensive and pre-hypertensive subjects.

The potent DPP-IV-inhibitory peptide, APFPE, was also quantified, and once again, the in vitro digestion of WPC fermented with RBC06 released the highest amount of this peptide, whereas unfermented WPC displayed the lowest concentration after digestion (Table 2).

## 4. Conclusions

In vitro GID of unfermented WPC and WPC fermented with selected *S. thermophilus* strains revealed that fermentation enhances the hydrolysis of WPC proteins. Since whey proteins, and especially β-lactoglobulin, are considered strong allergens, whey-based fermented beverages obtained with the selected *S. thermophilus* strains may improve the hydrolysis of these proteins, reducing their allergenicity and improving the health properties of these products. This is particularly evident when WPC was fermented with *S. thermophilus* strain RBC06, as the resulting hydrolysates produced a higher digestive hydrolysis of β-lactoglobulin with respect to WPC fermented with the other two strains of unfermented WPC. Furthermore, the increased WPC protein hydrolysis obtained by *S. thermophilus* fermentation could have a positive impact on the gastro-intestinal microbiota, as previously demonstrated for cheese digested samples [60,61]. Finally, WPC fermentation was pivotal to improve the biological activity profiles of digested samples, since WPC fermented with RBC06 displayed the highest ACE- and DPP-IV-inhibitory activities, whereas WPC fermented with RBN16 exhibited the highest antioxidant activity after in vitro digestion. Moreover, peptidomics analysis revealed a similar peptide profile between the different samples, but significant differences in the abundance of bioactive peptides. A direct relationship between relative amounts of ACE-inhibitory peptides and IC_50_ values against this enzyme strongly supports that the identified peptides were responsible for the observed bioactivity. Notably, in-vitro-digested WPC fermented with RBC06 showed the highest amount of bioactive peptides VPP, IPP, and APFPE.

In conclusion, this investigation provides a basis for the future exploitation of fermented WPC hydrolysates as possible functional beverages with improved benefits on human health. Significantly, WPC fermentation with *S. thermophilus* RBC06 promotes release during the digestion of quantities of VPP and IPP consistent with an in vivo effectiveness after a serving size consumption of this hydrolysate.

## Figures and Tables

**Figure 1 microorganisms-11-01742-f001:**
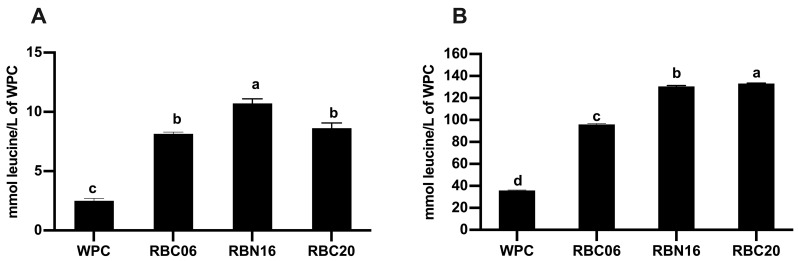
Assessment of protein hydrolysis before and after in vitro gastro-intestinal digestion. Analysis was carried out on undigested (**A**) and in-vitro-digested (**B**) unfermented or fermented whey protein concentrate (WPC). Fermentation was carried out with selected *S. thermophilus* strains, including RBC06, RBN16, and RBC20. Proteolytic activity was assessed using the TNBS assay after protein precipitation with trichloroacetic acid. The data are expressed as mmol of leucine equivalent/L of WPC. Values are the means of three assay replications ± standard deviation (SD). Different letters among the samples denote significant differences (*p* < 0.05).

**Figure 2 microorganisms-11-01742-f002:**
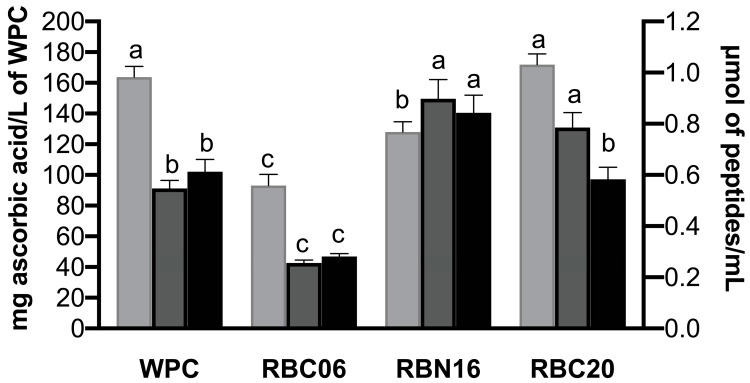
Biological activity profiles of in-vitro-digested low-molecular-weight peptide fractions of unfermented whey protein concentrate (WPC) and WPC fermented with *S. thermophilus* strains RBC06, RBN16, and RBC20. Low-molecular-weight peptide fractions were obtained through ultrafiltration at 10 kDa. Antioxidant activity (light grey bars) was assessed with the ABTS assay and the results are reported as mg of ascorbic acid equivalent per L of WPC (left *y*-axis). ACE-inhibitory (grey bars) and DPP-IV-inhibitory (black bars) activities were reported as IC_50_ values, defined as the sample concentration expressed in μmol of peptides/mL able to inhibit the enzyme activity by 50% (right *y*-axis). Values are the mean of three assay replications ±  standard deviation (SD). Different letters among the samples in the same assay denote significant differences (*p* < 0.05).

**Figure 3 microorganisms-11-01742-f003:**
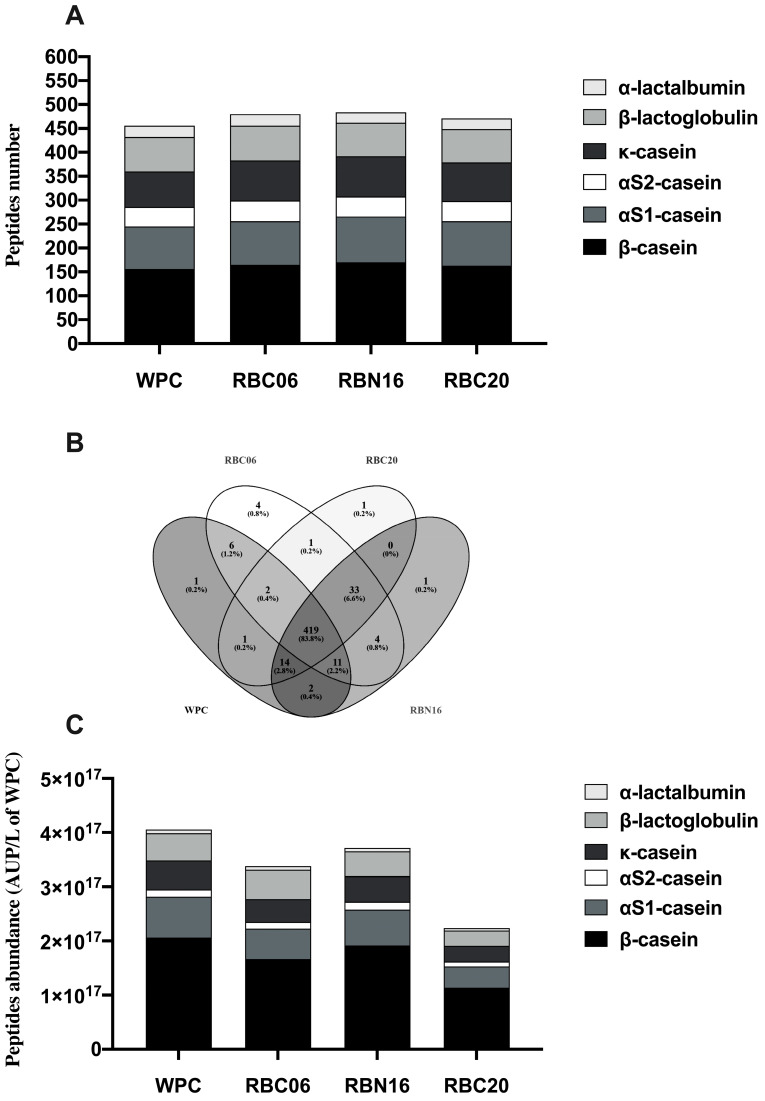
Peptidomics analysis of in-vitro-digested low-molecular-weight peptide fractions of the whey protein concentrate (WPC). Analysis was carried out on in-vitro-digested unfermented or WPC fermented by *S. thermophilus* strains RBC06, RBN16, and RBC20, respectively. Low-molecular-weight peptide fractions were obtained through ultrafiltration at 10 kDa. (**A**) Number of peptides identified in the WPC samples released from β-casein, αS1-casein, αS2-casein, κ-casein, β-lactoglobulin, and α-lactalbumin. (**B**) Venn diagram displaying differences among samples in the pattern of peptides released after in vitro digestion of WPC. (**C**) Peptide abundance for each protein in WPC samples. Data are reported as the sum of the intensity of each identified peptide measured as the area under the peak (AUP) through Skyline analysis. The complete list of identified peptides can be found in Appendix A.

**Figure 4 microorganisms-11-01742-f004:**
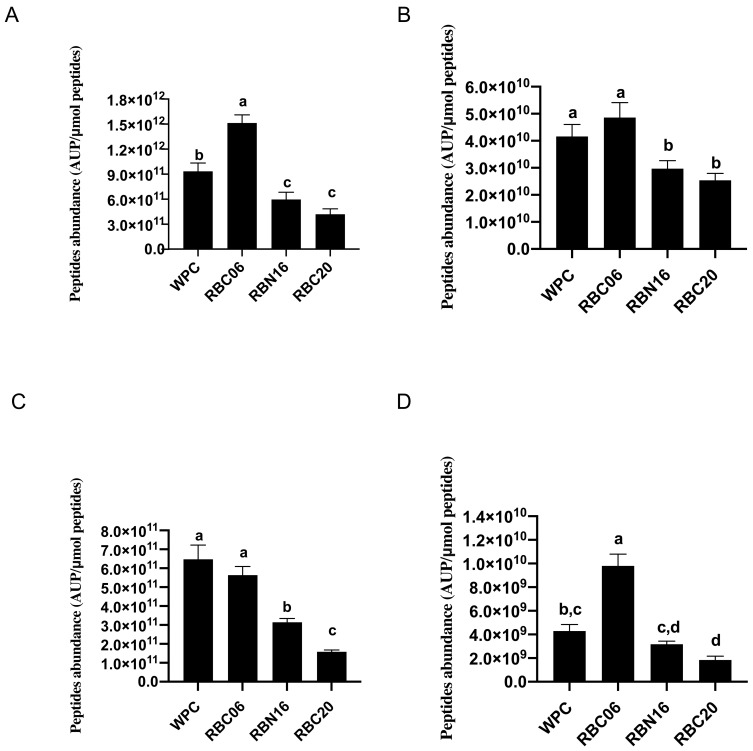
Peptidomics analysis of bioactive peptides identified in in-vitro-digested low-molecular-weight peptide fractions of unfermented whey protein concentrate (WPC) and WPC fermented with *S. thermophilus* strains RBC06, RBN16, and RBC20. Low-molecular-weight peptide fractions were obtained through ultrafiltration at 10 kDa. (**A**) Peptide abundance of ACE-inhibitory peptides in WPC samples. Data are reported as the sum of the intensity of each identified ACE-inhibitory peptide, measured as the area under the peak (AUP) via Skyline analysis, and normalized for the total peptide concentration in the specific sample (AUP/μmol of peptides). Different letters among the samples in the same assay denote significant differences (*p* < 0.05). (**B**) Peptide abundance of ACE-inhibitory peptides in WPC samples normalized for the respective IC_50_ value. Data are reported as the sum of the intensity of each identified ACE-inhibitory peptide, measured as the area under the peak (AUP) via Skyline analysis, and normalized for the peptide concentration in the specific sample (AUP/μmol of peptides). The AUP of each peptide was divided for the respective IC_50_ value. The complete list of the identified ACE-inhibitory peptides can be found in Table 1. Different letters among the samples in the same assay denote significant differences (*p* < 0.05). (**C**) Peptide abundance of DPP-IV-inhibitory peptides in WPC samples. Data are reported as the sum of the intensity of each identified DPP-IV-inhibitory peptide, measured as the area under the peak (AUP) via Skyline analysis, and normalized for the total peptide concentration in the specific sample (AUP/μmol of peptides). Different letters among the samples in the same assay denote significant differences (*p* < 0.05). (**D**) Peptide abundance of DPP-IV-inhibitory peptides in WPC samples normalized for the respective IC_50_ value. Data are reported as the sum of the intensity of each identified DPP-IV-inhibitory peptide, measured as the area under the peak (AUP) via Skyline analysis, and normalized for the peptide concentration in the specific sample (AUP/μmol of peptides). The AUP of each peptide was divided for the respective IC_50_ value. The complete list of the identified DPP-IV-inhibitory peptides can be found in Table 1. Different letters among the samples in the same assay denote significant differences (*p* < 0.05).

**Table 1 microorganisms-11-01742-t001:** Peptides sharing 100% of sequence homology with previously reported bioactive peptides identified within low-molecular-weight peptide fractions of in-vitro-digested unfermented whey protein concentrate (WPC) and WPC fermented with *S. thermophilus* strains RBC06, RBC20, and RBN16, respectively.

Peptide Sequence ^1^	Protein Fragment	Bioactivity ^2^	Sample
LVYPFPGPI	β-casein (58–66)	ACE inhibitor (IC_50_ = 180 μmol/L)	WPC, RBC06, RBC20, and RBN16
LVYPFP	β-casein (58–63)	ACE inhibitor (IC_50_ = 132 μmol/L)	RBC06, RBC20, and RBN16
VYPFPGPIPN	β-casein (59–68)	ACE inhibitor (IC_50_ = 325 μmol/L), antioxidant	WPC, RBC06, RBC20, and RBN16
VYPFPGPI	β-casein (59–66)	PEP inhibitor	WPC, RBC06, RBC20, and RBN16
YPFPGP	β-casein (60–65)	DPP-IV inhibitor (IC_50_ = 749 μmol/L), opioid	RBC06 and RBC20
YPFPGPI	β-casein (60–66)	ACE inhibitor (IC_50_ = 500 μmol/L), antioxidant, opioid, immunomodulator, anxiolytic, anti-cancer	WPC, RBC06, RBC20, and RBN16
YPFP	β-casein (60–63)	Opioid, anti-cancer	WPC, RBC06, RBC20, and RBN16
PFP	β-casein (61–63) αS1-casein (27–29)	ACE inhibitor (IC_50_ = 144 μmol/L)	RBC06 and RBN16
PFPGPIPN	β-casein (61–68)	ACE inhibitor	WPC, RBC06, RBC20, and RBN16
PFPGPI	β-casein (61–66)	Cathepsin B inhibitor	WPC, RBC06, RBC20, and RBN16
PGPIPN	β-casein (63–68)	ACE inhibitor, immunomodulator, anti-cancer, anti-inflammatory	WPC, RBC06, RBC20, and RBN16
SLPQ	β-casein (69–72)	ACE inhibitor (IC_50_ = 330 μmol/L)	WPC, RBC06, RBC20, and RBN16
PQNIPPL	β-casein (71–77)	DPP-IV inhibitor (IC_50_ = 1500 μmol/L)	WPC, RBC06, RBC20, and RBN16
IPP	β-casein (74–76) κ-casein (108–110)	ACE inhibitor (IC_50_ = 5 μmol/L), DPP-IV-inhibitor (IC_50_ = 169 μmol/L), antioxidant, anti-inflammatory	WPC, RBC06, RBC20, and RBN16
PVVVPPFLQPE	β-casein (81–91)	Anti-microbial	WPC and RBC06
VVPP	β-casein (83–86)	ACE inhibitor (IC_50_ = 258 μmol/L)	WPC, RBC06, RBC20, and RBN16
VPP	β-casein (84–86)	ACE inhibitor (IC_50_ = 8 μmol/L), antioxidant, anti-inflammatory	WPC, RBC06, RBC20, and RBN16
EAMAPK	β-casein (100–105)	Anti-microbial	WPC, RBC06, RBC20, and RBN16
EMPFPK	β-casein (108–113)	Anti-microbial	WPC, RBC06, RBC20, and RBN16
NLHLPLP	β-casein (132–138)	ACE inhibitor (IC_50_ = 51 μmol/L)	WPC, RBC06, RBC20, and RBN16
LHLP	β-casein (133–136)	ACE inhibitor (IC_50_ = 210 μmol/L)	WPC, RBC06, RBC20, and RBN16
LHLPLP	β-casein (133–138)	ACE inhibitor (IC_50_ = 3 μmol/L)	WPC, RBC06, RBC20, and RBN16
HLPLP	β-casein (134–138)	ACE inhibitor (IC_50_ = 41 μmol/L)	WPC, RBC06, RBC20, and RBN16
LPLPL	β-casein (135–139)	ACE inhibitor (IC_50_ = 325 μmol/L)	WPC, RBC06, RBC20, and RBN16
LPLP	β-casein (135–138)	ACE inhibitor (IC_50_ = 720 μmol/L)	WPC, RBC06, RBC20, and RBN16
PLP	β-casein (136–138)	ACE inhibitor (IC_50_ = 430 μmol/L)	WPC, RBC06, RBC20, and RBN16
KVLPVPQ	β-casein (169–175)	ACE inhibitor (IC_50_ = 1000 μmol/L), anti-inflammatory	WPC, RBC20, and RBN16
VLPVPQK	β-casein (169–175)	ACE inhibitor (IC_50_ = 15 μmol/L), antioxidant, anti-microbial	WPC, RBC06, RBC20, and RBN16
LPVPQ	β-casein (171–175)	DPP-IV inhibitor (IC_50_ = 44 μmol/L)	WPC, RBC06, RBC20, and RBN16
LPVP	β-casein (171–174)	DPP-IV inhibitor (IC_50_ = 87 μmol/L)	WPC, RBC06, RBC20, and RBN16
VPYPQ	β-casein (178–182)	DPP-IV inhibitor (IC_50_ = 41 μmol/L), antioxidant	WPC, RBC06, RBC20, and RBN16
PYPQ	β-casein (179–182)	Antioxidant	WPC, RBC06, RBC20, and RBN16
IQA	β-casein (187–189)	ACE inhibitor (IC_50_ = 33 μmol/L)	WPC, RBC06, RBC20, and RBN16
VLGP	β-casein (197–200)	ACE inhibitor (IC_50_ = 154 μmol/L), DPP-IV inhibitor (IC_50_ = 580 μmol/L)	WPC, RBC06, RBC20, and RBN16
VRGPFP	β-casein (201–206)	ACE inhibitor (IC_50_ = 592 μmol/L)	RBC06 and RBN16
APFPE	αS1-casein (26–30)	DPP-IV inhibitor (IC_50_ = 49 μmol/L)	WPC, RBC06, RBC20, and RBN16
YLG	αS1-casein (91–93)	Antioxidant	WPC and RBN16
PEL	αS1-casein (147–149)	Antioxidant	WPC, RBC06, RBC20, and RBN16
NPWDQ	αS2-casein (107–111)	Immunomodulator	WPC, RBC06, RBC20, and RBN16
VPITPT	αS2-casein (117–122)	DPP-IV inhibitor (IC_50_ = 130 μmol/L)	WPC, RBC06, RBC20, and RBN16
IPY	αS2-casein (201–203)	ACE inhibitor (IC_50_ = 206 μmol/L)	WPC, RBC06, RBC20, and RBN16
STVATL	κ-casein (141–146)	Anti-microbial	WPC, RBC06, RBC20, and RBN16
IPA	β-lactoglobulin (78–80)	ACE inhibitor (IC_50_ = 141 μmol/L), DPP-IV-inhibitor (IC_50_ = 49 μmol/L)	WPC, RBC06, RBC20, and RBN16
TPEVDDEALEK	β-lactoglobulin (125–135)	DPP-IV inhibitor (IC_50_ = 320 μmol/L)	WPC, RBC06, RBC20, and RBN16
ALPM	β-lactoglobulin (142–145)	ACE inhibitor (IC_50_ = 928 μmol/L)	WPC, RBC06, RBC20, and RBN16
YGG	α-lactalbumin (18–20)	Immunomodulator	WPC, RBC06, RBC20, and RBN16

^1^ IUPAC one letter code was used for describing the peptide amino acid sequence. ^2^ Peptide bioactivities and IC_50_ values were obtained from the Milk Bioactive Peptides Database. Abbreviations: ACE: angiotensin-converting enzyme; DPP-IV: dipeptidyl peptidase IV; PEP: prolyl-endopeptidase.

**Table 2 microorganisms-11-01742-t002:** Quantitative data of selected bioactive peptides in low-molecular-weight peptide fractions of in-vitro-digested unfermented whey protein concentrate (WPC) and WPC fermented with *S. thermophilus* strains RBC06, RBC20, and RBN16. Results are reported as mg/L of WPC.

Sequence	WPC	RBC06	RBC20	RBN16
IPP	3.00 ± 0.11 ^a^	40.57 ± 2.02 ^b^	11.86 ± 0.78 ^c^	12.13 ± 0.81 ^c^
VPP	1.68 ± 0.12 ^a^	18.55 ± 0.98 ^b^	3.35 ± 0.16 ^c^	4.72 ± 0.22 ^d^
APFPE	2.99 ± 0.20 ^a^	22.63 ± 1.25 ^b^	4.79 ± 0.14 ^c^	12.30 ± 0.93 ^d^

The peptides VPP and IPP are ACE-inhibitory and anti-hypertensive peptides, whereas the peptide APFPE is a DPP-IV-inhibitory peptide. Different letters in the same row indicate significantly different values (*p* < 0.05). An IUPAC one-letter code was used for describing the peptide amino acid sequence.

## Data Availability

The data presented in this study are available on request from the corresponding author.

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
