# Peer review of "Effect of Fermentation with Streptococcus thermophilus Strains on In Vitro Gastro-Intestinal Digestion of Whey Protein Concentrates"

_microorganisms, 2023, doi:10.3390/microorganisms11071742_

Round 1
Reviewer 1 Report
Dear authors and editors,
I extend my gratitude for your efforts and for providing me with the opportunity to review the manuscript. I have several comments that require attention in order to enhance the quality of the paper before considering it for publication in the Microorganisms journal. Additionally, I suggest further improvements to the writing style.
I kindly request the authors to make the following modifications and adhere to this structure for the abstract:
ü Background and problem: Clearly state the background information and the specific problem addressed in the study.
ü Rationale for the study: Provide a concise explanation of the reasons and justifications for conducting the research.
ü Research objectives: Clearly outline the objectives and goals of the study.
ü Methodology: Briefly describe the methodology employed in the research.
ü Important data and statistical analysis: Highlight the significant findings and include relevant statistical analysis, if applicable.
ü Conclusions: Summarize the key conclusions drawn from the study.
ü Novelty and importance of the findings: Emphasize the novelty and significance of the research outcomes.
ü Mention the three strains of Streptococcus thermophilus used in the study.
· The starter name (microorganism) should be in italics. Check and revise it in the whole manuscript.
· I would recommend the authors add more keywords with deep meaning.
· The acronyms or abbreviations should be defined the first time (and only one time); they appear in each of three sections: the abstract; the main text; the first figure or table (when defined for the first time, the acronym/abbreviation should be added in parentheses); then use acronyms. Check and revise it in the whole manuscript.
· The figures quality must be improved, and the authors must use one resolution, one font, one size, and so on with all the figures.
· The samples order in the figures must be started by WPC, then RBC06, RBN16, and RBC20. Check and modify all the figures.
· Figure 3 can be displayed in the vertical direction to make the figures clear; I could not see some data.
· Line 35 adds a reference.
· Line 37 adds a reference.
· In-vitro should be in italics, check and revise in the whole manuscript.
· Add one paragraph explaining the peptidomics, how they are coming, and what their types are. And how to analyse them (many methods)? Why did you select the UHPLC-MS/MS as the analyzing tool?
· Rewrite the aim of your work clearly with more details and how it is suitable for a food science major regarding food nutrition and the fermentation process.
· Justify novelty in Introduction and Discussion. The study has to be hypothesis-driven.
· In the chemicals section, the authors must mention all the chemicals and solvents they used in more detail and follow the journal format to write the chemicals information, their name, and the used mediums.
· Local producer, mention the company name, city, and country.
· Line 112 adds a reference for the preparation.
· Why did the authors store the reconstituted WPC at -20 °C? Why did they not use it directly in the experiments? How they use it after storage at -20 °C.
· Did the authors put the reconstituted WPC at 4 C for a few hours or a night?
· What about the reconstituted WPC condition? For such temperature, stirring, and how long?
· What about the safety of the strains? Did the authors study it? If they did it in a previous study, mention the reference.
· The fermentation conditions are not clear. What about the temperature, time, and concentration of the strain CFU/mL? Revise and add.
· Why did the authors select these conditions? Did they do pre-experiments? Or is there a reference? What is the reason for these selected parameters? Revise and add
· line 132, check the centrifugation condition. Do you mean xg?
· The section of whey protein concentrate preparation, fermentation, and in vitro gastro-intestinal digestion. It must be revised carefully and in detail.
· Line 144, check the centrifugation condition. Do you mean xg?
· Line 182, parameters described previously described???? Language?
· All Figures must be cited in the text and numbered by order of appearance. Check and revise.
· Do not start sentences with abbreviations or numbers. Check and revise.
· All Tables must be cited in the text and numbered by order of appearance.
· Tables must also stand alone and indicate the meaning of all abbreviations used on the table in a footnote. Also, mention the meaning of each letter in the peipted sequence LVYPFPGPI in the footnote. L = leucine,.....
· Line 299, avoid personal pronouns, such as we, they, you, I, or our , their, yours. Check and revise the whole manuscript.
· I would suggest the authors try to make the discussion in deep with the previous studies.
· Throughout the whole manuscript, the authors should write the scientific names of the bacteria in italic form. Please check and revise.
· The references section should be in complete journal format. Follow the author’s guidelines strictly.
The English need to be improved.
Reviewer 2 Report
The authors investigated the influence of Streptococcus fermentation on protein hydrolysis, bioactive peptides of whey protein concentrates (WPC) after gastro-intestinal digestion.
1. The title is misleading. The focus of the manuscript is not the effect of gastro-intestinal digestion but the effect of Streptococcus fermentation.
2. Some of the information in the discussion should be in the Introduction.
3. There is some repetition of statements, e.g. “Analysis was carried out on in vitro digested unfermented or fermented WPC. Fermentation was carried out with selected Streptococcus thermophilus strains RBC06, RBN16 and RBC20.”, which appear exactly the same in all Figure notes.
4. The authors must define all the acronyms whenever they first appear in the abstract or manuscript, e.g. angiotensin-converting enzyme ACE and dipeptidyl-peptidase-IV DPP-IV in the abstract.
5. Line 19: in the abstract, the sentence “In vitro gastro-intestinal digested WPC fermented with S. thermophilus RBN16 showed the highest antioxidant activity” contradicts with the manuscript. In Figure 2, it can be observed that WPC fermented with RBC20 has the highest radical scavenging activity. Similarly, in Line 282-3, it has the same result.
6. The authors should add more relevant information about biological fermentation and gastro-intestinal digestion to provide readers better background information upfront and more concise discussion of results.
7. Line 63: “However” should be “And”.
8. Line 67: delete “and”.
9. Line 83: isolate to isolated.
10. Line 113: Why “Reconstituted”.
11. Line 115-6: “previously selected for their ability to produce fermented WPC rich in bioactive peptides” should be “previously selected for their ability to ferment WPC rich in bioactive peptides”.
12. Line 120-1: At the end of each fermentation trial, each group of three replicates was pooled together. Besides, describe abbreviation INFOGEST.
13. Line 128-9: Do you mean “The further intestinal step of the digestion was initiated by adding 4 mL of intestinal fluid which could raise the pH to 7.5.”?
14. Line 136-7: I do not understand this sentence.
15. Line 146-7: There need to add some evidence to prove that it is credible to use leucine as an indicator of protein hydrolysis.
16. Line 228-235: these sentences will be better if they are in the introduction.
17. Line 288: The activity means the inhibitory ability.
18. Line 388: there has an empty page.
The authors should work a little more on the language. Some sentences have too much subordinate clauses so it may cause misunderstanding or miss critical information.
Round 2
Reviewer 1 Report
Accept